# The Relationship between Fat Mass Percentage and Glucose Metabolism in Children and Adolescents: A Systematic Review and Meta-Analysis

**DOI:** 10.3390/nu14112272

**Published:** 2022-05-28

**Authors:** Fangfang Chen, Junting Liu, Dongqing Hou, Tao Li, Yiren Chen, Zijun Liao, Lijun Wu

**Affiliations:** 1Department of Epidemiology, Capital Institute of Pediatrics, Beijing 100020, China; airechen@126.com (F.C.); yirenchen203@163.com (Y.C.); 2Child Health Big Data Research Center, Capital Institute of Pediatrics, Beijing 100020, China; winnerljt@163.com (J.L.); dqhou@sina.com (D.H.); socott@126.com (T.L.); 3Department of Integrated Early Childhood Development, Capital Institute of Pediatrics, Beijing 100020, China; zijun_liao@yeah.net

**Keywords:** fat mass percentage, children, glucose metabolism, systematic review, meta-analysis

## Abstract

To assess the relationship between fat mass percentage (FMP) and glucose metabolism in children aged 0–18 years we performed a systematic review of the literature on Medline/PubMed, SinoMed, Embase and Cochrane Library using the PRISMA 2020 guidelines up to 12 October 2021 for observational studies that assessed the relationship of FMP and glucose metabolism. Twenty studies with 18,576 individuals were included in the meta-analysis. The results showed that FMP was significantly associated with fasting plasma glucose (FPG) (*r* = 0.08, 95% confidence interval (CI): 0.04–0.13, *p* < 0.001), fasting plasma insulin (INS) (*r* = 0.48, 95% CI: 0.37–0.57, *p* < 0.001), and homeostasis model assessment (HOMA)- insulin resistance (IR) (*r* = 0.44, 95% CI: 0.33–0.53, *p* < 0.001). The subgroup analysis according to country or overweight and obesity indicated that these associations remained significant between FMP and INS or HOMA-IR. Our results demonstrated that there is a positive relationship between FMP and FPG. Moreover, subgroup analysis according to country or overweight and obesity indicated that FMP is significantly associated with INS and HOMA-IR. This is the first known systematic review and meta-analysis to determine the associations of FMP with glucose metabolism in children and adolescents.

## 1. Introduction

In 2021, approximately 537 million adults (20–79 years) are living with diabetes, which has become one of the most common metabolic disorders [1]. Type 2 diabetes (T2D) has been a global epidemic [2], and childhood obesity increases the risk of developing T2D [3]. Early prevention of childhood obesity may be critical for decreasing the risk of T2D [4].

Glucose metabolism disorders are frequently found among children with obesity [5], and adipose tissue is an underlying mechanism for the relationship between birth weight and T2D [6]. Significantly higher mean levels of glucose and insulin were observed among overweight and obese children than controls [7]. Meanwhile, the prevalence of prediabetes and diabetes was high among overweight and obese children [8]. Recently, it has been considered that the evaluation of body fat mass has a certain value in predicting abnormal glucose metabolism and metabolic syndrome in children [9,10]. Fat mass percentage (FMP) was positively associated with homeostasis model assessment insulin resistance (HOMA-IR) and glycosylated hemoglobin A1c (HbA1c), and South Asian children are more metabolically sensitive to adiposity [4]. It has been demonstrated that total and central adiposity are also positively associated with insulin resistance (IR) [11], which is a component of T2D and represents the most common metabolic disorder associated with obesity [12,13]. Abdominal adipose tissue is a powerful marker for inducing IR in obese children [14] and is a more significant predictor of metabolic syndrome traits in children than in adults [15]. Although fat mass is positively related to IR and T2D, the precise role of fat mass in glucose metabolism has not been fully elucidated.

Presently, body mass index (BMI) is commonly used to measure obesity, but it cannot comprehensively assess the type and location of obesity because it cannot differentiate muscle mass and fat mass.

Due to the important role of body fat mass in predicting glucose metabolism disorders and the lack of a meta-analysis assessing the relationship between FMP and glucose metabolism (blood glucose and IR) in children, the aim of the present study is to assess the relationship between FMP and glucose metabolism in children aged 0–18 years.

## 2. Materials and Methods

The protocol used for the systematic review is based on PRISMA 2020 (Preferred Reporting Items for Systematic reviews and Meta-Analyses) [16] and is registered in PROSPERO (International Prospective Register of Systematic Reviews; http://www.crd.york.ac.uk/prospero; accessed on 28 April 2020) under the registration number CRD42020150180.

The literature search was conducted using MEDLINE/PubMed, EMBASE, SinoMed and the Cochrane Library. We restricted the search to (1) studies published in English or Chinese and (2) studies with human (aged less than 18 years old) samples only.

Observational studies (cross-sectional or longitudinal design) relating FMP and glucose metabolism in children and adolescents were selected. To be included, studies were required to meet the following criteria: (1) FMP was measured by the direct method, BIA/DXA/ADP/the isotopic dilution method/the four-compartment (4C) model of the body composition method, etc.; (2) assessed at least one glucose metabolism indicator (fasting glucose, IR, HOMA-IR, or HbA1c.

Studies were not included if (a) the study outcomes only focused on surrogate indicators of body composition or physical fitness rather than directly measured indicators; (b) they assessed the relationship in adults or nonhumans; (c) they involved children with severe physical limitations; (d) they referred to children with disorders that could limit generalizability (mitochondrial dysfunction, Prader–Willi syndrome, nonalcoholic fatty liver disease, polycystic ovarian syndrome, and mental disorders including attention-deficit/hyperactivity disorder (ADHD), conduct or neuropsychiatric disorders including schizophrenia, or any detected delay in communication, adaptive, cognition or socioemotional domains); and (e) they referred to a particular population group, such as an aboriginal group, immigrant group, economic status or nonhealthy child.

### 2.1. Data Extraction (Selection and Coding)

The abstracts of the retrieved studies were screened by two review team members, Fangfang Chen and Junting Liu. Any disagreements over the data collection were resolved by consensus, with a third researcher, Dongqing Hou, being asked if consensus could not be reached. Studies that provided the association between FMP and glucose metabolism in children and adolescents were selected and recorded. The indices of glucose metabolism included blood glucose, insulin, HbA1c, and HOMA-IR. The association described as correlation coefficient (*r*) and regression coefficient (*β*) and odds ratio (OR) values in the studies will be extracted and recorded as the main data. Other data will be extracted, including author, publication year, study year, study location, study design, body composition assessment method, sample size, age, and gender of the study population.

### 2.2. Risk of Bias (Quality) Assessment

‘Agency for Healthcare Research and Quality (AHRQ, Rockville, MD, USA)’ quality assessment forms recommended for cross-sectional/prevalence studies, which have 11-item checklists, were used to assess the methodological quality of the selected studies independently by two researchers (Fangfang Chen and Junting Liu). Any disagreements over the assessment of the risk of bias will be discussed, and a consensus will be reached, with a third researcher (Dongqing Hou) being consulted if consensus cannot be reached.

An item was scored ‘0’ if it was answered ‘NO’ or ‘UNCLEAR’; if it was answered ‘YES’, then the item scored ‘1’. Article quality was assessed as follows: low quality = 0–3; moderate quality = 4–7; and high quality = 8–11.

### 2.3. Strategy for Data Synthesis

The amount the observed variance reflecting real differences in the effect size across the included trials was graded with the *I*^2^ statistic, with values representing mild, moderate, and severe heterogeneity (<25%, 25–75%, and >75%, respectively) [17]. Subgroup analysis clustered by gender was processed according to the results of the heterogeneity test and the quality of the studies. The correlation coefficient (*r*) and regression coefficient (*β*) values of FMP and glucose metabolism (fasting plasma glucose (FPG), fasting plasma insulin (INS), and HOMA-IR) were synthesized. HOMA-IR = (INS (μU/mL) * FPG (mmol/L))/22.5. OR values were also extracted when variables were categorized and OR values were provided. Tables were created to summarize the characteristics of the selected studies, and the feasibility of conducting a meta-analysis was determined after the data were extracted.

R software was used to combine the pooled data. A fixed-effect model was used if there was no evidence of heterogeneity; otherwise, a random-effects model was used.

### 2.4. Analysis of Subgroups or Subsets

Subgroup and meta-regression analyses were performed with respect to the main factors causing heterogeneity, such as the measurement of FMP in different studies, the gender of the study participants, the year of study, the study country, and the study population. Furthermore, the design and methodological quality of the studies were considered for additional subgroup analyses.

### 2.5. Analysis of Publication Bias

We used funnel plots to assess publication bias, when ≥10 studies were included in the meta-analysis. Asymmetry of the funnel plot was measured via Egger’s test, with a *p* value < 0.05 indicating statistical significance.

## 3. Results

### 3.1. Identification and Selection of Studies

The original search identified 4423 studies. After removal of duplicates and elimination of papers based on eligibility criteria, 20 studies remained (Figure 1). Of the 20 studies included, 11 studies assessed the relationship between FMP and FPG, 14 studies assessed it between FMP and INS, 17 studies focused on the relationship between FMP and HOMA-IR, and 1 study calculated the correlation coefficient (*r*) between FMP and HbA1c.

### 3.2. Study Characteristics

The characteristics of all 20 studies [4,18,19,20,21,22,23,24,25,26,27,28,29,30,31,32,33,34,35,36] are shown in Table 1. The papers were based on cross-sectional study or cohort study, and were conducted in France, South Africa, the U.S.A., Guatemala, Brazil, England, China, Malaysia, Mexico, and Italy. Table 2 shows the quality evaluation of the methodology according to AHRQ quality assessment forms (US 2004). In the case of AHRQ scoring, there were 1 study with 10 points, 1 study with 9 points, 1 study with 8 points, 3 studies with 7 points, 5 studies with 6 points, 5 studies with 5 points, 3 studies with 4 points, and 1 study with 2 points. There were 3 high-quality trials with an average score of 9 points, 16 moderate-quality trials with an average score of 5.5 points, and 1 low-quality trial with 2 points.

### 3.3. Relationship between FMP and FPG

Referring to the relationship between FMP and FPG, seven studies provided the correlation coefficient (*r*) (study ID: 1–7) (Figure 2A). Parrett et al. (study ID: 1) and Zheng et al. (study ID: 2) reported that FMP was negatively associated with FPG, but other studies (study ID: 3–7) reported that FMP was positively associated with FPG. The data of 7 studies (*n* = 2002) were pooled, and the results showed that after data pooling using a fixed-effect model, FMP was significantly associated with FPG (Figure 2A) (*r* = 0.08, 95% confidence interval (CI): 0.04–0.13, *p* < 0.001). There was stratified analysis of the relationship between FMP and FPG in 3 studies (study ID: 2, 6, 7). After data pooling using a fixed-effect model (Figure 2B), FMP was significantly associated with FPG in females (*r* = 0.10, 95% CI: 0.03–0.18, *p* = 0.006), and there was no significant association between FMP and FPG in males (*r* = 0.06, 95% CI: −0.01–0.14, *p* = 0.09).

Three studies also gave the regression coefficient (*β*) through linear analysis (study ID: 8, 9, 10) (Table 1). Coutinho et al. (study ID: 8) reported that FMP was positively associated with FPG (*β* = 0.09). Nightingale et al. (study ID: 10) also reported the positive relationship between FMP and FPG (*β* were expressed in 1 standard deviation (SD, 1SD = 9.1) increase in FMP; *β* = 0.69, 95% CI: 0.38–1.01, in males; *β* = 0.41, 95% CI: 0.09–0.73, in females). Faria et al. (study ID: 9) reported consistent results (BMI-age, P25–P75: *β* = 0.25; BMI-age, >P85: *β* = −0.22).

### 3.4. Relationship between FMP and INS

Referring to the relationship between FMP and INS, eleven studies provided the correlation coefficient (*r*) (study ID: 1–7, 12, 14–16) (Figure 3). The data of 11 studies (*n* = 2460) were pooled (heterogeneity, *I*^2^ = 86%), and the results showed that after data pooling using a random-effect model, FMP was significantly associated with fasting plasma insulin levels (Figure 3A) (*r* = 0.48, 95% CI: 0.37–0.57, *p* < 0.001).

After excluding each study in turn, it was found that after removing the study with the greatest heterogeneity (study ID: 6), the heterogeneity slightly decreased but remained at a high level. Since no significant reduction in heterogeneity was observed, we explored the source of study heterogeneity. The heterogeneity was assessed using the *I*^2^ statistic, and the results showed that compared with age, the measurement of FMP in different studies, as well as the year of study, the study country and the study population might explain sources of heterogeneity better. When stratified by country, after data pooling using a fixed-effect model (study ID: 1, 5, 6, 12, 16; heterogeneity, *I*^2^ = 0%), FMP was significantly associated with INS in the American population (Figure 3B) (*r* = 0.59, 95% CI: 0.55–0.62, *p* < 0.001). When stratified by overweight and obesity, after data pooling using a fixed-effect model (study ID: 2, 3, 14; heterogeneity, *I*^2^ = 18%), we also found that FMP was significantly associated with INS (Figure 3C) (*r* = 0.35, 95% CI: 0.25–0.44, *p* < 0.001).

Four studies also gave the regression coefficient (*β*) through linear analysis (study ID: 2, 8, 9, 13) (Table 1). Zheng et al. (study ID: 2) reported that FMP was positively associated with INS (*β* = 0.149, in males; *β* = 0.226, in females). Coutinho et al. (study ID: 8, *β* = 0.38), Faria et al. (study ID: 9, BMI-age P25–P75: *β* = 0.003; BMI-age > P85: *β* = 0.009), and Ouyang et al. (study ID: 13, *β* = 0.16, in males; *β* = 0.14, in females) reported consistent results.

### 3.5. Relationship between FMP and HOMA-IR

Referring to the relationship between FMP and HOMA-IR, 11 studies provided the correlation coefficient (*r*) (study ID: 1–7, 14–17) (Figure 4). The data of 11 studies (*n* = 2409) were pooled (heterogeneity, *I*^2^ = 86%), and the results showed that after data pooling using a random-effect model, FMP was significantly associated with HOMA-IR (Figure 4A) (*r* = 0.44, 95% CI: 0.33–0.53, *p* < 0.001).

After excluding each study in turn, it was found that after removing one study with the greatest heterogeneity (study ID: 6), the heterogeneity slightly decreased but remained at a high level. Since no significant reduction in heterogeneity was observed, we explored the source of study heterogeneity. The heterogeneity was assessed using the *I*^2^ statistic, and the results showed that compared with age, the measurement of FMP in different studies, as well as the year of study, the study country, and the study population might explain sources of heterogeneity better. When stratified by country, after data pooling using a fixed-effect model (study ID: 1, 5, 6, 16; heterogeneity, *I*^2^ = 35%), FMP was significantly associated with HOMA-IR in the American population (Figure 4B) (*r* = 0.57, 95% CI: 0.53–0.60, *p* < 0.001). When stratified by overweight and obesity, after data pooling using a fixed-effect model (study ID: 2, 3, 14, 17; heterogeneity, *I*^2^ = 0%), we also found that FMP was significantly associated with HOMA-IR (Figure 4C) (*r* = 0.33, 95% CI: 0.24–0.42, *p* < 0.001).

Seven studies also gave the regression coefficient (β) through linear analysis (study ID: 2, 8–10, 18–20) (Table 1). Among these studies, six studies (study ID: 2, 8–10, 18, 19) reported that FMP was positively associated with HOMA-IR, and one study (study ID: 20) reported that FMP was negatively associated with HOMA-IR.

In addition, Chen et al. (study ID: 11) reported that FMP was positively associated with impaired fasting glucose (IFG) (IFG: FPG ≥ 5.6 mmol/L; ORs were expressed in 1 SD increase in FMP; OR = 1.18, 95% CI: 1.11–1.26, in males; OR = 1.17, 95% CI: 1.09–1.26, in females). Nightingale et al. (study ID: 10) reported the positive relationship between FMP and HbA1c (*β* were expressed in 1 SD (1SD = 9.1) increase in FMP; *β* = 4.07, 95% CI: 2.77–5.36, in males; *β* = 2.65, 95% CI: 1.34–3.96, in females).

### 3.6. Publication Bias Evaluation

When ≥10 studies were included in the meta-analysis, publication bias was assessed using funnel plots and Egger’s test. The results showed that the distribution of included studies on both sides of the funnel plot was asymmetric. A publication bias was detected between FMP and INS or HOMA-IR (Appendix A), which was mainly derived from between-research heterogeneity.

## 4. Discussion

Considering that fat mass plays a critical role in glucose metabolism, most previous research concerns the positive relationship of fat mass with IR and T2D. However, the precise role of fat mass in glucose metabolism has not been fully elucidated. There has been no meta-analysis to assess the relationship between FMP and glucose metabolism in children.

To exclude the influence of confounding factors, such as age and degenerative disease, the study of the pediatric population could be better to clarify the relationship between FMP and glucose metabolism. Our research is the first meta-analysis of the relationship between FMP and glucose metabolism in children and adolescents. The results demonstrated that FMP is positively associated with FPG in children and adolescents. Moreover, subgroup analysis according to country or overweight and obesity indicated that FMP is significantly associated with INS and HOMA-IR.

Interestingly, after subgroup analysis stratified by gender, the results showed that FMP was significantly associated with FPG in females, but there was no significant association between FMP and FPG in males. This suggests that there might be gender differences in the relationship between FMP and FPG. Muscles have estrogen receptors, which can increase the rate of glucose uptake into the muscle when they are activated. It is well known that skeletal muscle is an important component of body composition and a metabolic sink for glucose disposal [37]. There are also differences between genders in where fat is stored and the characteristics of that fat. These factors cause differences in metabolism between men and women. A recent study reported that the positive correlation between FMP and insulin in girls was more pronounced than that in boys [38]. This might be one of the reasons why there were gender differences in the relationship between FMP and FPG.

Skeletal muscle also plays an important role in the regulation of blood glucose levels and affects the development of IR and T2D [39]. A cross-sectional study revealed that trunk muscle quality was related to glucose tolerance [40], and a longitudinal study showed a positive association between abdominal muscle quantity and T2D [41]. A previous study reported that decreased skeletal muscle mass is associated with deterioration of insulin sensitivity [42]. Thus, targeting the mechanism(s) underlying skeletal muscle activity in glucose metabolism may prevent or delay IR and T2D.

Our literature search was conducted using MEDLINE/PubMed, EMBASE, SinoMed, and the Cochrane Library, and the original search identified 4423 studies. After removal of duplicates and elimination of papers based on eligibility criteria, 20 studies remained. Of the 20 studies included, 11 studies (study ID: 1–11) assessed the relationship between FMP and FPG, 14 studies (ID: 1–9, 12–16) assessed it between FMP and INS, and 17 studies (study ID: 1–10, 14–20) focused on the relationship between FMP and HOMA-IR. There were some studies of the associations of FMP with FPG, INS, and HOMA-IR in adult populations. For example, Borschmann et al. reported that reducing sedentary time and fat mass within 6 months of stroke might improve glucose tolerance and insulin resistance [43]. Zeng et al. showed that high fat mass increased fasting glucose, HOMA-IR, triglycerides, decreased high-density lipoprotein cholesterol, and high fat-free mass reduced HOMA-IR, triglycerides, and low-density lipoprotein cholesterol [44]. Park et al. reported that appendicular fat (%) had a negative correlation with glucose, log insulin, and HbA1c [45]. Müller et al. showed that small decreases and increases in fat mass were associated with corresponding decreases and increases in insulin secretion as well as increases and decreases in insulin sensitivity [46]. However, these studies could not exclude the influence of confounding factors such as age and degenerative disease.

Childhood obesity increases the risk of abnormal glucose metabolism. Anthropometric measures, such as BMI and waist-to-height ratio, are widely used to evaluate obesity, but high levels of these indicators do not equal high FMP. BMI and waist-to-height ratio are widely used to evaluate adiposity owing to their feasibility and low cost, but their use is limited because they do not discriminate for muscle mass and fat mass and cannot comprehensively assess the type and location of obesity [47]. A higher BMI may arise not only from greater body fat, but also from either higher lean mass, bone mass, or both, making it an imperfect measure of adiposity [48,49]. The evaluation of body fat mass has a certain value in predicting abnormal glucose metabolism and metabolic syndrome in children. A recent study also suggests that the associations of body fat and bone parameters with insulin were clearer when FMP was used instead of BMI to classify obesity [38]. Since FMP plays an important role in glucose metabolism, it is important to study the relationship between FMP and glucose metabolism in children.

Previous studies showed that sex hormones are significantly correlated with glucose and lipid metabolism. Premenopausal women exhibit increased insulin sensitivity compared with men, but this advantage disappears after menopause, in part owing to a reduction of 17β-estradiol (E2). Xu et al. reported that serum E2 and sex hormone binding globulin (SHBG) could influence HbA1c levels in non-diabetic populations [50]. Kim et al. showed that E2, testosterone, and SHBG levels are associated with lipid subfractions in adults [51]. Similar studies have been conducted in children. Adolescent glucose metabolism may be influenced by testosterone, perhaps partially via skeletal muscle mass [52]. Total testosterone was negatively associated with fasting glucose, insulin, and HOMA-IR, but these associations were attenuated by additional adjustment for skeletal mass index or FMP.

There was a strong relationship between insulin and SHBG, and SHBG could be a marker for either hyperinsulinemia, IR, or both, in obese children (aged 6–9 years) [53]. Ortega et al. also showed that SHBG is related to insulin and HOMA-IR independently of age in both sexes [54]. A recent study reported that E2 was negatively correlated with leptin in normal-weight boys, whereas testosterone was negatively correlated with leptin in overweight/obese boys [55].

Sex hormone levels and SHBG also have important effects on high-density lipoprotein (HDL) and IR among children and adolescents [56]. E2 drives a typically atheroprotective lipid profile through upregulation of HDL/ApoA1 [57]. SHBG levels are related to a decrease in HDL and ApoA1 levels during puberty in boys and to a decrease in triglycerides levels during puberty in both sexes [58].

Obesity is known to impact reproductive hormones during puberty. Youth with obesity have lower SHBG than those with normal weight [59]. During weight loss, SHBG and insulin sensitivity increased significantly in obese children, and the changes were independent of gender, puberty, and the changes in BMI [60].

However, several limitations should also be addressed. First, due to limited conditions, only MEDLINE/PubMed, EMBASE, SinoMed, and the Cochrane Library were searched, and only studies published in English or Chinese were included. There have been limited studies on FMP and glucose metabolism in children and adolescents, and there might be omissions. Second, in this study, funnel plot analysis was performed on FMP and INS or HOMA-IR. The results showed that the distribution of included studies on both sides of the funnel plot was asymmetric, indicating that there was potential publication bias in FMP and INS or HOMA-IR, which might be mainly derived from study heterogeneity. Nevertheless, our findings could avoid some potential bias because the results of stratification according to confounders, such as the study country and the study population, showed significant associations of FMP with INS and HOMA-IR. Third, heterogeneity among the studies was significant, which could be partly explained by the study country and the study population. Fourth, this is a systematic literature review examining the association between FMP and glucose metabolism through observational studies. The search selection criteria for research design included cross-sectional studies, and most of the articles actually searched are cross-sectional studies. In statistical analysis, correlation analysis was also attempted. However, meta-analysis data on correlation using cross-sectional studies have limitations in identifying causation.

## 5. Conclusions

From this systematic review and meta-analysis, we demonstrated that there is a positive relationship between FMP and FPG. Moreover, subgroup analysis according to country or overweight and obesity indicated that FMP is significantly associated with INS and HOMA-IR. This is the first known systematic review and meta-analysis to determine the associations of FMP with glucose metabolism in children and adolescents. Our results provide novel information to fill a gap in the literature. Further examination of these relationships requires more cohort studies.

## Figures and Tables

**Figure 1 nutrients-14-02272-f001:**
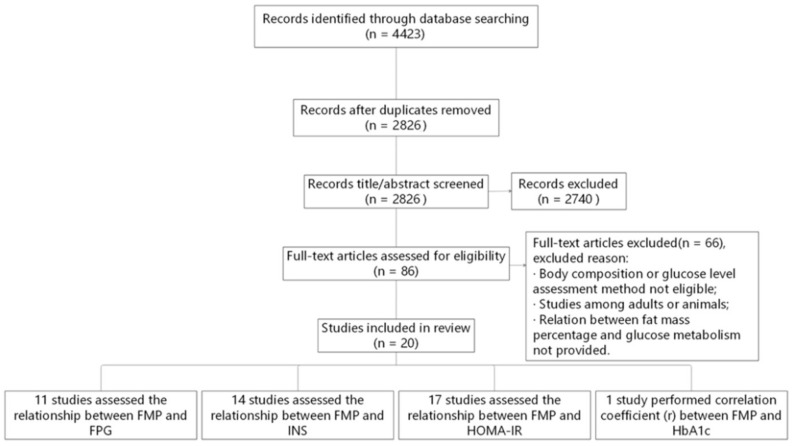
Flow diagram of the literature search and paper selection process. FMP, fat mass percentage; FPG, fasting plasma glucose; HbA1c, glycosylated hemoglobin A1c; HOMA-IR, homeostasis model assessment insulin resistance; INS, fasting plasma insulin.

**Figure 2 nutrients-14-02272-f002:**
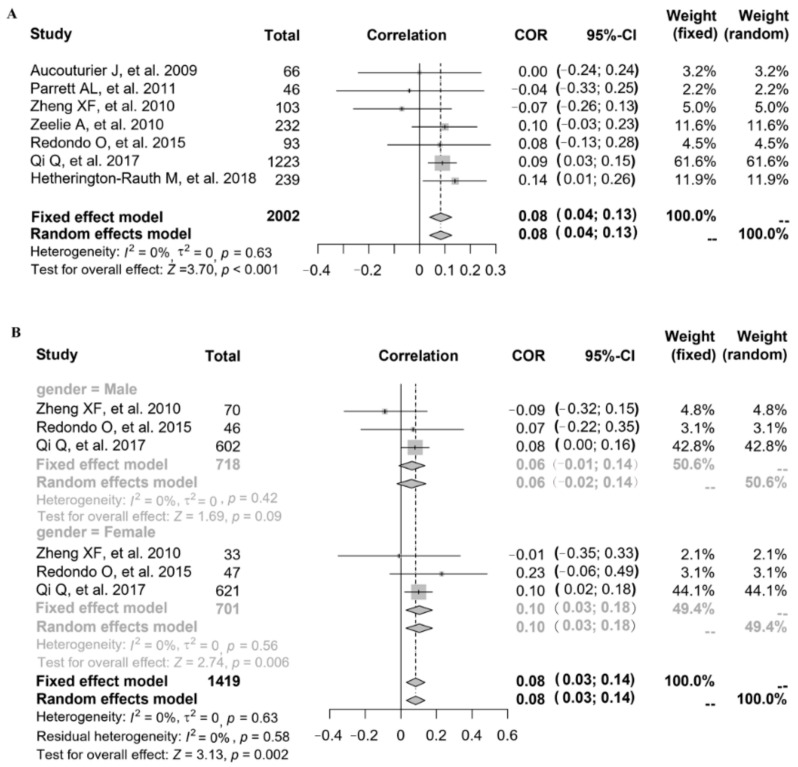
The relationship between FMP and FPG: (**A**) 7 studies (study ID: 1–7) [18,19,20,21,22,23,24] provided the correlation coefficient (*r*); (**B**) stratified by gender in 3 studies (study ID: 2, 6, 7) [19,23,24].

**Figure 3 nutrients-14-02272-f003:**
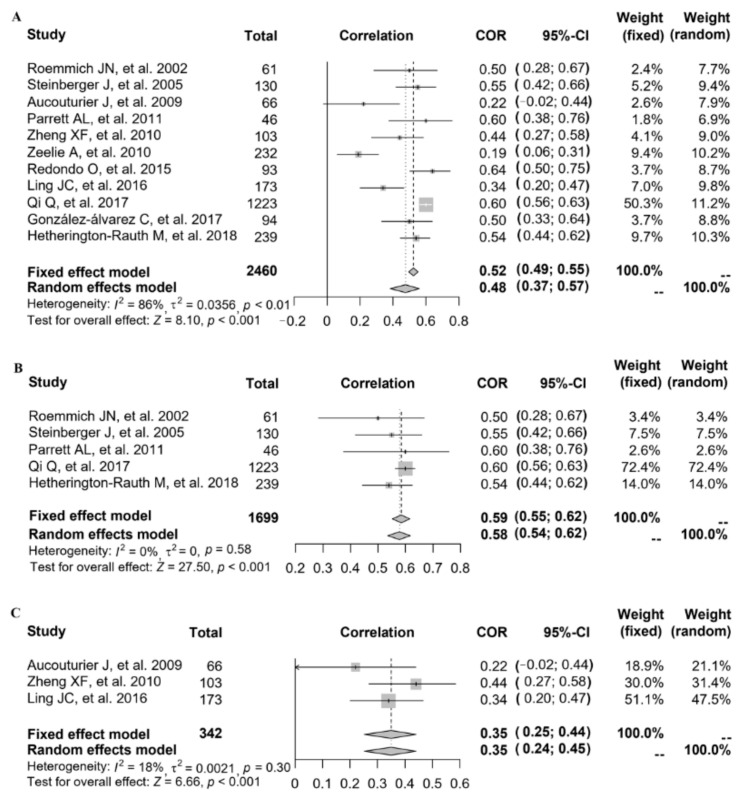
The relationship between FMP and INS: (**A**) 11 studies (study ID: 1–7, 12, 14–16) [18,19,20,21,22,23,24,28,30,31,32] provided the correlation coefficient (*r*); (**B**) stratified by countries in 5 studies (study ID: 1, 5, 6, 12, 16) [18,22,23,28,32]; (**C**) stratified by overweight and obesity in 3 studies (study ID: 2, 3, 14) [19,20,30].

**Figure 4 nutrients-14-02272-f004:**
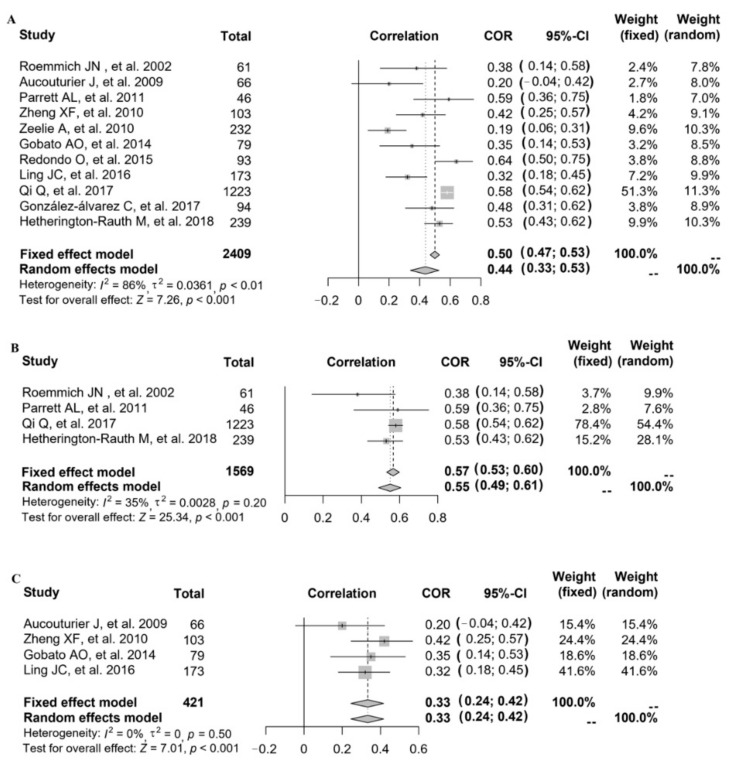
The relationship between FMP and HOMA-IR: (**A**) 11 studies (study ID: 1–7, 14–17) [18,19,20,21,22,23,24,30,31,32,33] provided the correlation coefficient (*r*); (**B**) stratified by countries in 4 studies (study ID: 1, 5, 6, 16) [18,22,23,32]; (**C**) stratified by overweight and obesity in 4 studies.

**Table 1 nutrients-14-02272-t001:** Characteristics of the included trials.

Study ID	Reference ID	Study	Study Year	Study Location (s)	Study Design	Sample Size (Total, M/F)	Age (Years)	Body Composition Assessment	Which Bio-Marker Related with FMP Were Reported	Indices of Correlation *
1	[18]	Parrett AL, et al. 2011	not clear	USA	cross sectional study	46, 24/22	7–12	DXA	FPG, INS, HOMA-IR	*r*
2	[19]	Zheng XF, et al.2010	2007–2009	not clear	cross-sectional study	103, 70/33	7–18	BIA	FPG, INS, HOMA-IR	*r*, *β*
3	[20]	Aucouturier J, et al. 2009	2005–2007	France	cross-sectional study	66, 35/31	children and adolescents	DXA	FPG, INS, HOMA-IR	*r*
4	[21]	Zeelie A, et al. 2010	2003	a low socio-economic area in the North-WestProvince of South Africa	cross-sectional study	232, 99/133	15–19	ADP	FPG, INS, HOMA-IR	*r*
5	[22]	Hetherington-Rauth M, et al. 2018	not clear	Tucson, Arizona, USA. Hispanic	cross-sectional study	239, F	9–12	DXA	FPG, INS, HOMA-IR	*r*
6	[23]	Qi Q, et al. 2017	2011	USA	cross-sectional study	1223, 602/621	8–16	BIA	FPG, INS, HOMA-IR	*r*
7	[24]	Redondo O, et al. 2015	not clear	Guatemala	cross-sectional study	93, 46/47	7–12	DXA	FPG, INS, HOMA-IR	*r*
8	[25]	Coutinho PR, et al. 2015	2013	Curitiba Paraná, Brazil	cross-sectional study	53, F	13–17	DXA	FPG, INS, HOMA-IR	*β*
9	[26]	Faria FR, et al. 2013	2010	Vicosa, Minas Gerais, Brazil	cross-sectional study	210, 100/110	15–18	BIA	FPG, INS, HOMA-IR	*β*
10	[4]	Nightingale CM, et al. 2013	August 2004–February 2007	England	cross-sectional study	4633, 2237/2396	9–10	BIA	FPG, HOMA-IR, HbA1c	*β*
11	[27]	Chen F, et al. 2019	2013–2015	Beijing, China	cross sectional study	7926, 4036/3890	6–17	DXA	FPG	OR ^§^
12	[28]	Steinberger J, et al. 2005	not clear	Minnesota, USA	cross-sectional study	130, 72/58	11–17	DXA	INS	*r*
13	[29]	Ouyang F, et al. 2010	1998–2000	Anqingregion, Anhui Province, China	cohort study; use the follow up data	1613, 888/725 ^#^	13–20	DXA	INS	*β*
14	[30]	Ling JC, et al. 2016	2012	Kuala Lumpur, Malaysia	cross-sectional study	173, 53/120	12.9 ± 0.4	BIA	INS, HOMA-IR	*r*
15	[31]	González-Álvarez C, et al. 2017	not clear	Mexican	cross-sectional study	94, 44/50	5–11	DXA	INS, HOMA-IR	*r*
16	[32]	Roemmich JN, et al. 2002	not clear	USA	cross-sectional study	61, 30/31	prepuberty to late puberty	the four-compartment (4C) model of body composition	INS, HOMA-IR	*r*
17	[33]	Gobato AO, et al. 2014	2008–2009	Brazil	cross-sectional study	79, 40/39	10–18	DXA	HOMA-IR	*r*
18	[34]	Barbosa-Cortes L, et al. 2015	not clear	San Mateo Capulhuac, Mexico City	cohort study; use the follow up data	41, 15/26	<12	the isotopic dilutionmethod	HOMA-IR	*β*
19	[35]	Santos LC, et al. 2008	2004	São Paulo, Brazil	cross-sectional study	49, 12/37	16.6 ± 1.4	DXA	HOMA-IR	*β*
20	[36]	Bedogni G, et al. 2012	2004–2009	Piancavallo, Verbania, Italy	cross-sectional study	1512, 628/884	6–18	BIA	HOMA-IR	*β*

M: male; F: female; DXA: dual energy X-ray absorptiometry; BIA: bioelectrical impedance analysis; ADP: air-displacement plethysmography; FMP: fat mass percentage; FPG: fasting plasma glucose; INS: insulin; HOMA-IR, homeostasis model assessment insulin resistance; HbA1c: glycosylated hemoglobin A1c. * *r* values are displayed and merged in figures. ^#^ Twins were involved in this study. ^§^ Impaired fasting glucose as outcome variable, logistic regression were used in risk markers for 1 SD increase in adiposity measured, and ORs were all expressed.

**Table 2 nutrients-14-02272-t002:** The quality evaluation of the methodology according to Agency for Healthcare Research and Quality (AHRQ, US 2004) assessment forms *.

Study ID	Reference ID	Inclusion Research	AHRQ 11-Item Checklist											Total Score	Total Score
			A	B	C	D ^#^	E	F	G	H	I	J	K		
1	[18]	Parrett AL, et al. 2011	1	0	0	1	0	1	0	1	0	0	0	4	moderate quality
2	[19]	Zheng XF, et al. 2010	0	1	1	1	0	1	0	1	0	0	0	5	moderate quality
3	[20]	Aucouturier J, et al. 2009	1	0	0	1	0	0	0	0	0	0	0	2	low quality
4	[21]	Zeelie A, et al. 2010	1	0	1	1	0	1	0	1	0	0	0	5	moderate quality
5	[22]	Hetherington-Rauth M, et al. 2018	1	1	0	1	1	1	1	1	0	0	0	7	moderate quality
6	[23]	Qi Q, et al. 2017	1	1	1	1	0	1	1	1	0	0	0	7	moderate quality
7	[24]	Redondo O, et al. 2015	1	1	0	1	0	1	0	1	0	0	0	5	moderate quality
8	[25]	Coutinho PR, et al. 2015	1	1	1	1	1	1	1	1	0	1	0	9	high quality
9	[26]	Faria FR, et al. 2013	1	1	1	1	0	1	0	1	0	0	0	6	moderate quality
10	[4]	Nightingale CM, et al. 2013	1	1	1	1	0	1	0	1	0	1	0	7	moderate quality
11	[27]	Chen F, et al. 2019	1	1	1	1	0	1	1	1	0	1	0	8	high quality
12	[28]	Steinberger J, et al. 2005	1	1	0	1	0	1	0	1	0	0	0	5	moderate quality
13	[29]	Ouyang F, et al. 2010	1	1	1	1	0	1	1	1	1	1	1	10	high quality
14	[30]	Ling JC, et al. 2016	1	1	0	1	0	1	0	0	0	1	0	5	moderate quality
15	[31]	González-Álvarez C, et al. 2017	1	1	0	1	0	1	0	0	0	0	0	4	moderate quality
16	[32]	Roemmich JN, et al. 2002	0	0	0	1	1	1	0	1	0	0	0	4	moderate quality
17	[33]	Gobato AO, et al. 2014	1	1	1	1	0	1	0	0	0	1	0	6	moderate quality
18	[34]	Barbosa-Cortes L, et al. 2015	1	0	0	1	0	1	0	1	0	1	1	6	moderate quality
19	[35]	Santos LC, et al. 2008	1	1	0	1	0	1	1	1	0	0	0	6	moderate quality
20	[36]	Bedogni G, et al. 2012	1	1	1	1	0	1	0	1	0	0	0	6	moderate quality

* 1 = yes; 0 = no or unclear; ^#^ Study 1–8 were all population-based. A, define information source; B, list inclusion/exclusion criteria; C, indicate identifying patients time; D, indicate subjects consecutive or population-based; E, subjective components masked; F, quality assessments; G, patients exclusions explained; H, confounding either assessed, controlled, or both; I, handle missing data; J, summarize data completeness; K, clarify follow-up. Article quality was assessed according to the total score as follows: low quality = 0–3; moderate quality = 4–7; high quality = 8–11.

## Data Availability

The data presented in this study are available on request from the corresponding author.

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
