# Peer review of "The Relationship between Fat Mass Percentage and Glucose Metabolism in Children and Adolescents: A Systematic Review and Meta-Analysis"

_nutrients, 2022, doi:10.3390/nu14112272_

Round 1

Reviewer 1 Report

Thank you for the opportunity to review a manuscript entitled “The Relationship between Fat Mass Percentage and Glucose Metabolism in Children and Adolescents: A Systematic Review and Meta-Analysis”. The manuscript presents a meta-analysis of the available literature on the relationship between directly measured fat mass and fasting plasma/insulin levels or HOMA-IR. It confirms a positive association between fat mass and glucose metabolism markers.

Overall, the manuscript is well written. The work is well described in all necessary details. The topic is correctly introduced, methods are correctly chosen, and the findings are clearly presented and sufficiently discussed.

I can see a few minor points to improve:

Introduction

Line 50 – “in the current study” does it fit properly into the context?

Lines 53 – 56: It would be better to clearly define the study aim at this place. In this form, it is rather a conclusion

Discussion

The discussion could confront the study with more references. For example, paragraph 269-273 could be significantly expanded. The abbreviation WHtR is needless if not repeatedly used.

Best regards.

Author Response

Reviewer 1:

Question 1. Line 50 – “in the current study” does it fit properly into the context?

Answer: The reviewer is very careful. Thank you very much! We revised the sentence in the introduction (Line 51), as follows:

Presently, body mass index (BMI) is commonly used to measure obesity, but it cannot comprehensively assess the type and location of obesity because it cannot differentiate muscle mass and fat mass.

Question 2. Lines 53 – 56: It would be better to clearly define the study aim at this place. In this form, it is rather a conclusion

Answer: We agree with the reviewer’s comment. Thank you very much! We revised the sentence in the introduction (Line 56), as follows:

the aim of the present study is to assess the relationship between FMP and glucose metabolism in children aged 0-18 years.

Question 3. The discussion could confront the study with more references. For example, paragraph 269-273 could be significantly expanded. The abbreviation WHtR is needless if not repeatedly used.

Answer: We agree with the reviewer’s comment. Thank you very much! We deleted the abbreviation WHtR. We added some sentences and references in the discussion (Line 279-295), as follows:

Our literature search was conducted using MEDLINE/PubMed, EMBASE, Si-noMed and the Cochrane Library, and the original search identified 4423 studies. After removal of duplicates and elimination of papers based on eligibility criteria, 20 studies remained. Of the 20 studies included, 11 studies (study ID: 1-11) assessed the relation-ship between FMP and FPG, 14 studies (ID: 1-9, 12-16) assessed it between FMP and INS, and 17 studies (study ID: 1-10, 14-20) focused on the relationship between FMP and HOMA-IR. There were some studies of the associations of FMP with FPG, INS, and HOMA-IR in adult population. For example, Borschmann et al reported that reducing sedentary time and fat mass within 6 months of stroke might improve glucose tolerance and insulin resistance [37]. Zeng et al showed that high fat mass increased fasting glucose, HOMA-IR, triglycerides, decreased high-density lipoprotein cholesterol, and high fat-free mass reduced HOMA-IR, triglycerides, and low-density lipoprotein cholesterol [38]. Park et al reported that appendicular fat (%) had a negative correlation with glucose, log insulin, and HbA1c [39]. Müller et al showed that small decreases and increases in fat mass were associated with corresponding decreases and increases in insulin secretion as well as increases and decreases in insulin sensitivity [40]. However, these studies could not exclude the influence of confounding factors such as age and degenerative disease.

Reviewer 2 Report

In this manuscript titled, " The Relationship between Fat Mass Percentage and Glucose Metabolism in Children and Adolescents: A Systematic Review and Meta-Analysis.", Fangfang Chen et al., authors focused on assessing the relationship between fat mass percentage (FMP) and glucose metabolism in 13 children aged 0-18 yrs. This manuscript is written clearly, however, the manuscript appears preliminary.

 1.     In the introduction, the authors introduced the condition of  T2D in 2014. How about the recent 5 years? Authors should update the background.

Author Response

Reviewer 2:

Question 1. In the introduction, the authors introduced the condition of T2D in 2014. How about the recent 5 years? Authors should update the background.

Answer: The reviewer is very careful. We agree with the reviewer’s comment. Thank you very much! We have updated the reference 1 and 3, and revised the sentence in the introduction (Line 30-31), as follows:

In 2021, approximately 537 million adults (20-79 years) are living with diabetes, which has become one of the most common metabolic disorders [1].